# SCALE-ADAPTIVE DIFFUSION MODEL FOR COMPLEX SKETCH SYNTHESIS

**Jijin Hu[1]**       **Ke Li[1]** *       **Yonggang Qi[1]**       **Yi-Zhe Song[2]**
[1]Beijing University of Posts and Telecommunications, CN       [2]SketchX, CVSSP, University of Surrey, UK
{jijinhu,like1990,qiyg}@bupt.edu.cn       y.song@surrey.ac.uk

## ABSTRACT

While diffusion models have revolutionized generative AI, their application to human sketch generation, especially in the creation of complex yet concise and recognizable sketches, remains largely unexplored. Existing efforts have primarily focused on vector-based sketches, limiting their ability to handle intricate sketch data. This paper introduces an innovative extension of diffusion models to pixel-level sketch generation, addressing the challenge of dynamically optimizing the guidance scale for classifier-guided diffusion. Our approach achieves a delicate balance between recognizability and complexity in generated sketches through scale-adaptive classifier-guided diffusion models, a scaling indicator, and the concept of a residual sketch. We also propose a three-phase sampling strategy to enhance sketch diversity and quality. Experiments on the QuickDraw dataset showcase the potential of diffusion models to push the boundaries of sketch generation, particularly in complex scenarios unattainable by vector-based methods.

## 1 INTRODUCTION

The field of diffusion models (Ho et al., 2020; Song et al., 2020a;b; Dhariwal & Nichol, 2021; Ho & Salimans, 2021) has seen remarkable progress, pushing the boundaries of generative AI and enabling the generation of high-quality images across diverse domains (Meng et al., 2021; Choi et al., 2021; Rombach et al., 2022; Poole et al., 2022; Wang et al., 2023). However, this surge of advancements has largely overlooked the unique challenge posed by human sketch generation—a task demanding the creation of complex sketches that maintain a delicate balance between conciseness and recognizability. Recent endeavors in this direction have predominantly centered on vector-based sketches (Ha & Eck, 2018; Chen et al., 2017; Zang et al., 2021). Unfortunately, these vector-based approaches, while suitable for simpler sketches, grapple with inherent limitations when tackling intricate and complex sketch data (Das et al., 2022; Wang et al., 2022).

In this paper, we embark on an ambitious endeavor to harness the full potential of diffusion models by extending their capabilities into the realm of pixel-based sketch generation. Our overarching goal is to demonstrate their prowess in generating complex sketches that strike the perfect balance—concise yet highly recognizable. Determining the appropriate level of complexity in sketch generation has long been considered a formidable challenge, primarily due to the inherent variability in line structures within sketches. To address this challenge, we adapt a conventional classifier-guided pipeline (Dhariwal & Nichol, 2021) designed specifically for sketch generation. However, this transition is not without its difficulties, as we quickly encounter a significant hurdle: the conventional designs, meticulously fine-tuned for photo generation, do not seamlessly transfer to the intricate realm of sketch creation (as illustrated in Figure 1).

As we delve deeper into this problem space, we uncover a compelling revelation—one that diverges from the well-established principles governing photography. In the realm of photos, higher scale values often correlate with increased fidelity (Dhariwal & Nichol, 2021; Ho & Salimans, 2021). However, in the context of sketch generation, we encounter a fascinating phenomenon that we term "over-sketching" (as depicted in Figure 1 (a)). When working with larger scale values, we witness the emergence of repetitive strokes, overlaying previously rendered lines and ultimately compromising the quality of the generated sketches. Addressing this issue proves challenging, as there exists

---

*Correspondence to: Ke Li (like1990@bupt.edu.cn). Code to be found at GitHub page

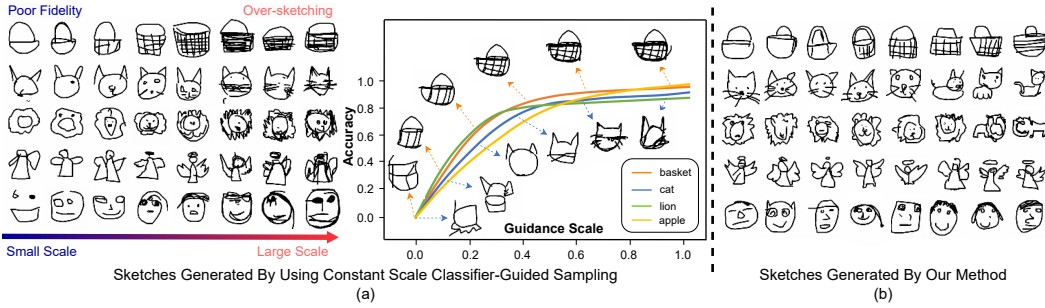

Figure 1: (a) Vanilla constant scale classifier guided sketch sampling suffers from either insufficient recognizability or over-sketching. In addition, there is no universal scale that is suitable for all categories. (b) Sketches generated by our model are highly recognizable and more visually appealing.

no universal scale choice suitable for diverse sketch categories. Moreover, adopting smaller scales may result in insufficient recognizability, a trade-off that is also undesirable.

Our paramount contribution, therefore, revolves around the development of a specialized classifier-guided diffusion model meticulously crafted for the domain of sketches. At its core, our model introduces a dynamic sampling scale selector. This selector grapples with the intricate task of determining the optimal scale for each distinct sketch while ingeniously sidestepping the issue of over-sketching. This delicate equilibrium ensures that our generated sketches strike the perfect harmony between recognizability and complexity.

This strategy pivots around two integral components that work in tandem: a scaling indicator and a residual sketch. The residual sketch offers us a nuanced perspective on the influence of classifier guidance by tracking how the generated sketch evolves, pixel by pixel, under varying scale choices. This empowers us to pinpoint the scale where the residual sketch best aligns with the scaling indicator, thereby optimizing the entire generation process.

To further elevate the quality of our generated sketches, we introduce two supplementary design elements. Firstly, we demonstrate the effectiveness of commencing the generation process with a few unconditional sampling steps. This initial phase allows the rough structure of the sketch to take form, amplifying the diversity of the generated sketches by maximizing mode coverage. Secondly, we address the gradual attenuation of classifier gradients as the sampling deepens. To counteract this, we strategically implement an early-stop mechanism within our scale adaptive sampling. This seamless transition back to unconditional denoising accelerates the process while concurrently refining the generation results by eliminating noisy pixels.

Our contributions can be summarized as follows: (i) We introduce scale adaptive classifier-guided diffusion models tailored for pixel-based sketch generation, replacing the conventional fixed gradient scale approach and achieving high-quality sketch generation. (ii) We present a novel scaling indicator that optimizes classifier guidance based on recognizability and complexity, complemented by the innovative concept of the residual sketch, enabling fine-tuned control of the generation process in raw pixel space for improved sketch quality. (iii) Our three-phase sampling strategy, comprising shape and structure construction, scale adaptive sampling for class-specific sketches, and denoising, significantly enhances sample diversity and quality by removing background clutter. These contributions collectively advance the state-of-the-art in sketch generation with diffusion models.

## 2 RELATED WORK

**Sketch Generation.** Synthesizing human sketches is an appealing task that increasingly received attention in recent years. Early studies (Song et al., 2018; Wang et al., 2018; Li et al., 2019b;a; Yu et al., 2020) and more recent arts (Chan et al., 2022; Wan et al., 2022) focused on the problem of image-to-sketch generation, to help understand and mimic how humans perceive and represent the visual world using sketches. Another line of work is however concentrating on how to better capture the sequential features in human sketches within the single domain, involving RNN-based (Ha & Eck, 2018; Su et al., 2020; Chen et al., 2017), GAN-based (Ge et al., 2020; Liu et al., 2019), and Graph-based (Xu et al., 2021; Yang et al., 2021b) approaches. These models typically adopt

a sequence decoder, i.e., LSTM or Transformer, to explicitly capture the geometric structure of the sequential points represented in coordinates or parametric Bézier curve (Das et al., 2020). As a result, the sketch generation is formed as an autoregressive process. Most recently, diffusion models (Das et al., 2022; Wang et al., 2022) are leveraged to directly learn the distribution of points' coordinates in a non-autoregressive manner, thereby advancing in generating complex sketches. Instead of using the sequential representation of stroke points, we seek to train diffusion models on the raster sketches composed of pixel grids to generate high-quality sketches. An additional property of pixel-based diffusion modeling is that the classifier gradients (Dhariwal & Nichol, 2021) can be conveniently applied as guidance without retraining the unconditional diffusion models or extra differentiable rasterization rendering.

**Guided Diffusion Models.** There is a large body of literature on controllable generation using diffusion models. The pioneering work ADM (Dhariwal & Nichol, 2021) allows image generation conditioned on a class label by adding the classifier gradients to the frozen unconditional trained diffusion. Later, a Classifier-free approach (Ho & Salimans, 2021) is importantly proposed to avoid separately training the classifier while achieving similar sample quality, thereby triggering plenty of work on text-conditional image synthesis, e.g., Stable Diffusion (Rombach et al., 2022), DALL-E 2 (Ramesh et al., 2022), GLIDE (Nichol et al., 2022) and Imagen (Saharia et al., 2022) to name a few. More broadly, latest works expand the scope of conditions to different modalities via cross-attention or adapter with CLIP, such as segmentation mask (Gafni et al., 2022), sketch (Voynov et al., 2023), and many others (Zhang & Agrawala, 2023; Mou et al., 2023). Our work is different from previous works in that the strength of the classifier guidance is dynamically determined to manage the line complexity, to improve the realism of produced sketches.

## 3 BACKGROUND

On a high level, diffusion models can sample data from a simple Gaussian distribution by reversing noisy data gradually in multiple steps. It typically consists of two inverse processes, i.e., the forward for diffusion and the backward for denoising.

**Diffusion and Denoising** The forward process is a predefined diffusion process $q$, which gradually adds Gaussian noise to a real image $x_0$, resulting noisier versions $x_{1:T}$. It is formally defined as $q(x_t|x_{t-1}) = \mathcal{N}(x_t; \sqrt{1 - \beta_t}x_{t-1}, \beta_t\mathbf{I})$, where $0 < \beta_t < 1, t = 1 \dots T$ is a predefined variance schedule to specify the noise levels of $x_t$. The backward process is a denoising function $p_\theta$, where a neural network is trained to produce slightly clearer data $x_{t-1}$ from $x_t$ at each timestep. Given $p_\theta$, we can sample from pure noise $x_T$ and sequentially produce samples $x_{T-1}$, $x_{T-2}$, $\dots$ until reaching $x_0$, i.e., a produced sample.

**Learning Objective** As the backward process is also formulated as a Gaussian, i.e., $p_\theta(x_{t-1}|x_t) = \mathcal{N}(x_{t-1}; \mu_\theta(x_t, t), \Sigma_\theta(x_t, t))$, where the mean and variance of the Gaussian are parameterized by $\mu_\theta$ and $\Sigma_\theta$, respectively. DDPM (Ho et al., 2020) show that the variance can be set to time-dependent constant, i.e., $\Sigma_\theta(x_t, t) = \sigma_t^2\mathbf{I}$, the mean $\mu_\theta(x_t, t)$ is then reparametrized by a noise approximator $\epsilon_\theta$ since we have $\mu_\theta(x_t, t) = 1/\sqrt{\alpha_t} \cdot \left(x_t - \beta_t/\sqrt{1 - \bar{\alpha}_t}\epsilon_\theta(x_t, t)\right)$, where $\alpha_t = 1 - \beta_t$ and $\bar{\alpha}_t := \prod_{s=1}^{t} \alpha_s$. Consequently, an alternative training objective $||\epsilon_\theta(x_t, t) - \epsilon||^2$, i.e., MSE loss between the true and the estimated noise, is derived for training the diffusion models.

**Classifier-guided Sampling** To generate data conditioned on class labels, a classifer $p_\phi(y|x_t)$ can be trained on noisy data $x_t$. Then the gradient of the classifier, i.e., $\nabla_{x_t} \log p_\phi(y|x_t)$, is leveraged to guide the sampling. Specifically, the predicted noise after classifier guidance is:

$$\hat{\epsilon} = \epsilon_\theta(\boldsymbol{x}_t, t) - s \cdot \sqrt{1 - \bar{\alpha}_t}\nabla_{\boldsymbol{x}_t} \log p_\phi(y|\boldsymbol{x}_t) \tag{1}$$

where $s$ is a gradient scale predefined manually. Then the class conditional sampling is achieved by replacing $\epsilon_\theta$ with $\hat{\epsilon}$. It turns out that the scaling factor $s$ has a significant impact on the generated data, and increasing $s$ will typically trade off the diversity for fidelity (Ho & Salimans, 2021).

## 4 METHODOLOGY

Our goal is to generate new sketches in pixel format conditioned on class labels using the learned denoising function, i.e., $p_\theta(x_{t-1}|x_t, y)$, using DDIM sampler (see Appendix A for more details

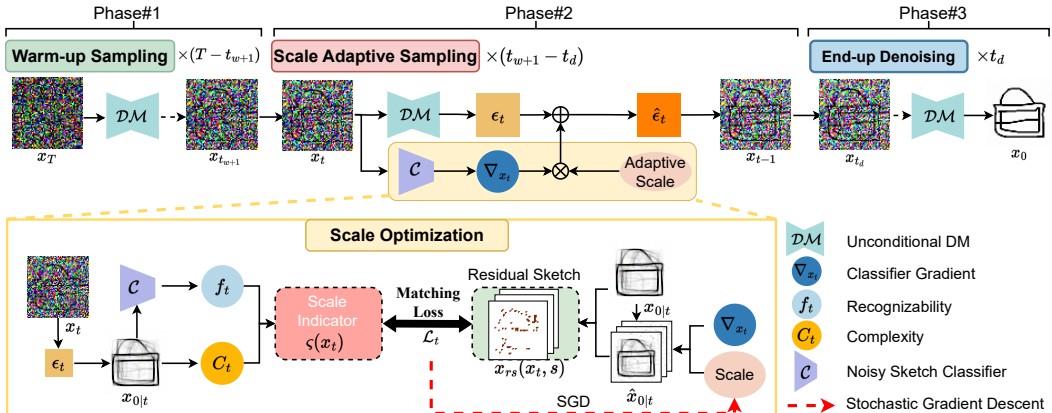

Figure 2: Schematic overview of our pixel-level sketch generator. There are three sampling phases, i.e., warm-up sampling (phase #1), scale adaptive classifier-guided sampling (phase #2), and end-up denoising (phase #3). Core to our framework is phase #2 which can adaptively select an optimal classifier guidance scale $s$ to encourage better recognizability and avoid over-sketching, thereby boosting the sample quality. Essentially, the scale is dynamically determined by matching a scale indicator and residual sketches at each sampling step. The scale indicator is to signal the demand for classifier guidance by predicting the final generation results $x_{0|t}$. The residual sketch measures whether the chosen scale could activate proper guidance as the indicator suggests.

about DDIM sampling) under the guidance of classifier gradients as described above. Particularly, an adaptive scaling strategy is devised to dynamically determine the level of the gradient scale at each time step to improve the quality of the produced sketches. A schematic overview of our model is shown in Figure 2. Details are described in the following.

## 4.1 SCALE ADAPTIVE CLASSIFIER-GUIDED SAMPLING

Following the standard pipeline for handling natural images, we first train a DDPM using sketch images, i.e., $x_0 \in \mathbb{R}^{H \times W \times 3}$. During generation, as shown in Figure 1, a constant value of the scale $s$ in Eq. (1) will lead to sub-optimal quality of the produced sketches, suffering from either insufficient recognizability or over-sketching. Therefore, a scale adaptive sampling strategy is developed to overcome the above issues. Specifically, for each intermediate sampling step $t$, a scaling indicator is used to penalize the guidance strength when the complexity and recognizability of the expected produced sketch $x_{0|t}$ are already sufficiently high. A residual sketch image $x_{rs}$ is then to explicitly measure the impact of any gradient scale, by leveraging the visual difference of $x_{0|t}$ before and after performing the guidance. Then the scale is optimized by using a differentiable matching module to encourage the residual sketch to conform to the scaling indicator, thereby steering the generation accordingly. In the following, we will formally define each key module.

**Scaling Indicator.**     High recognizability and proper complexity are two crucial properties to make the produced sketches informative and visually appealing. Therefore, we empirically define a scaling indicator by combining these two factors:

$$\varsigma(x_t) = \gamma \cdot \exp(\alpha \cdot (1 - c(x_{0|t})) - \beta \cdot f(x_{0|t})), \qquad (2)$$

where $x_{0|t} = (x_t - \sqrt{1 - \bar{\alpha}_t} \cdot \epsilon_\theta(x_t, t))/\sqrt{\bar{\alpha}_t}$, $c(x_{0|t})$ and $f(x_{0|t})$ are all scalar, denoting the complexity and the recognizability of an expected produced sketch $x_{0|t}$, respectively. Specifically, the stroke complexity $c(x_{0|t})$ is heuristically defined as the fraction of stroke pixels to the whole canvas: $c(x_{0|t}) = \frac{1}{HW} \sum_{HW} ||x_{0|t}||_0$. The recognizability $f(x_{0|t})$ is given by the probability of the estimated produced sketch $x_{0|t}$ being classified to the conditional class $y$, i.e., $f(x_{0|t}) = p_\phi(y|x_{0|t})$, where the classifier $p_\phi$ is parameterized by $\phi$, and trained using noisy sketches. Intuitively, the scale indicator $\varsigma(x_t)$ is to signal the demand for classifier guidance. For example, a high level of either complexity or recognizability will derive a small $\varsigma(x_t)$, suggesting a stop sign for applying the classifier guidance. In contrast, a large $\varsigma(x_t)$ implies the classifier guidance in need. $\alpha$ and $\beta$

are used to balance the effects between $c(x_{0|t})$ and $f(x_{0|t})$. In the following, we will show how to optimize the scale $s$ according to the scaling indicator by introducing residual sketch.

**Residual Sketch.** To measure the impact of classifier guidance on $x_{0|t}$, i.e., the estimated final-step sketch $x_0$ based on the current sample $x_t$, we can compare two versions of $x_{0|t}$ before and after performing the classifier guidance, i.e., $x_{0|t}$ and $\hat{x}_{0|t}$. Namely, a residual sketch $x_{rs}$ is developed to represent the per-pixel differences between $x_{0|t}$ and $\hat{x}_{0|t}$. Formally, $x_{rs}$ is defined as follows:

$$x_{rs}(x_t, s) = \left| M(\hat{x}_{0|t}) - M(x_{0|t}) \right| \tag{3}$$

where $M(\cdot) \in \mathbb{R}^{H \times W}$ is a `Sigmoid` function to transform a sketch (pixel values are firstly averaged across the RGB channels) into a soft binary mask, i.e., most of the pixel values are projected near zeros or ones. And $|\cdot|$ is to ensure the entries in $x_{rs} \in \mathbb{R}^{H \times W}$ are all positive.

**Scale Optimization.** Here, we show how to optimize the gradient scale $s$ according to the scaling indicator at each sampling step. Basically, this is achieved by enforcing the residual sketch $x_{rs}$ to be synchronized with the scale indicator $\varsigma(x_t)$. Intuitively, $x_{rs}$ should be an empty mask if $\varsigma(x_t)$ suggests a stop sign for applying the classifier guidance. Otherwise, $x_{rs}$ should be richly painted if $\varsigma(x_t)$ is large, indicating guidance in high demand. Therefore, an optimization objective can be formulated as follows:

$$L_t(s) = \frac{1}{2} \sum_{i=1}^{N} \left( \varsigma(x_t^{(i)}) - \frac{1}{HW} \sum_{HW} x_{rs}(x_t^{(i)}, s) \right)^2 \tag{4}$$

where $N$ is the number of sketches generated within a sampling batch, and $L_t(s)$ is the mean squared error between the scaling indicator $\varsigma(x_t)$ and the global average pooling of the residual sketch $x_{rs}(x_t, s)$. Stochastic gradient descent (SGD) is employed to obtain the optimal value of $s$ at each timestep $t$ by minimizing $L_t(s)$.

## 4.2 WARM-UP SAMPLING FOR DIVERSITY EXPANSION

Prior works Dhariwal & Nichol (2021); Ho & Salimans (2021) reveal that increasing the strength of the classification guidance can improve the sample precision (i.e., fidelity), while at the cost of recall, i.e., the degraded diversity of the generation. We show that beginning with a few unconditional sampling steps as warm-up can considerably alleviate the issue, i.e., boost the diversity of the generated samples. An empirical principle is applied to determine how many unconditional sampling steps are conducted for the warm-up. The idea is to carry out unconditional generation until the overall structure has been shaped. To achieve the goal, we can simply measure if the classification probability of the top-1 class, i.e., $p(c_{1st}|x_{0|t})$, exceeds any other classes by a pre-defined margin $\eta$. Therefore, the end step $t_w$ of the warm-up sampling can be set by:

$$t_w = t, \quad \text{if } p_\phi(c_{1st}|x_{0|t}) - p_\phi(c_{2nd}|x_{0|t}) > \eta \tag{5}$$

## 4.3 END-UP UNCONDITIONAL DENOISING

We observe that as the sampling goes on, the classifier gradient will gradually vanish and almost has no effect afterward. Therefore, we early-stop the classifier-guided sampling when the expected $x_{0|t}$ at timestep $t_d$ is sufficiently recognizable, i.e.,

$$t_d = t, \quad \text{if } p_\phi(y|x_{0|t}) > \xi \tag{6}$$

where $\xi$ is a threshold to determine the endpoint of the guidance. However, it turns out that the produced sketches are still noisy (i.e., lots of clutter pixels scattered across the image) once the classifier-guided sampling is ceased at the cut-off point $t_d$. We find out that continuing to progress unconditional denoising till the end (i.e., the pre-defined total number of DDIM sampling steps) will eventually produce sketches with clean backgrounds.

## 5 EXPERIMENTS

### 5.1 EXPERIMENTAL SETUP

**Dataset.** The current largest doodle dataset *QuickDraw*, which has 345 common object categories, is adopted for model training and evaluation. In our experiments, a small subset, i.e., 30

categories[1] are first randomly chosen to facilitate a thorough yet easier comparison with other baseline methods. Meanwhile, our model is trained on the complete 345 classes and compared with a few top-notch generative models to validate the scalability of our approach.

**Competitors.** There are two categories of baseline methods based on the representation of the sketches, i.e., vector-based or raster-based. Vectorized sketch generation competitors include SketchRNN (Ha & Eck, 2018), SketchHealer (Su et al., 2020), SketchAA (Yang et al., 2021a), SketchKnitter (Wang et al., 2022), and ChiroDiff (Das et al., 2022). Due to the lack of existing raster-based sketch generation approaches, three strong image generation models, i.e., StyleGAN2 (Karras et al., 2020), DDIM [2](Song et al., 2020a), and classifier-free diffusion guidance (CFDG[3])(Ho & Salimans, 2021), are employed as alternatives for comparison.

**Evaluation metrics.** Several standard metrics, including FID, precision, and recall, are leveraged for evaluation. Fréchet Inception Distance (FID) is widely used to measure the fidelity of the RGB images produced by generative models. To tailor it as a reasonable measurement for sketches, the same network Inception-V3 is employed but further finetuned on *QuickDraw* dataset for classification. Then the obtained customized Inception-V3 is utilized as a feature extractor, which is used to calculate the distance between the generated samples and the real data. Precision and recall are typically adopted by diffusion models to validate the quality and mode coverage of the generated samples. We follow (Nichol & Dhariwal, 2021) to employ the improved precision and recall metrics (Kynkäänniemi et al., 2019) to assess the generation results.

*Language Aligned Expressivity.* We additionally proposed to utilize CLIP-Score (Radford et al., 2021) to measure the expressiveness of the generated sketches. Intuitively, the produced sketches would express similar visual concepts to real sketches. In practice, CLIP-Score is leveraged to measure the distance between the generated sketches and the text descriptions of real ones, where the text descriptions are sourced by manually summarizing the visual content in the *QuickDraw* dataset. More specifically, five captions are constructed for each category under the template "*this is a sketch of* X", where X is either a coarse or fine-grained linguistic caption, e.g., X = "a girl's face with two pigtails". A full list of the captions is attached in Appendix E. Then a pre-trained CLIP model ViT-L/14 (Dosovitskiy et al., 2021) is used to extract the features of the generated sketches and the pre-defined descriptions using the image encoder and text encoder, respectively. The CLIP-Score is then defined as the averaged similarity between the generated sketches and their closest captions. Given CLIP-Score, we further propose a CLIP-Fine score which measures whether the retrieved top-1 captions are fine-grained or not.

**Implementation details.** The same U-Net proposed in ADM (Dhariwal & Nichol, 2021) is employed as the noise predictor, and 10k sketches per category (batch size is 64) in the training set are used to train our model for 200k iterations. The default size of the produced sketches is set to $64 \times 64$. Four Nvidia 3090 GPUs are used and the learning rate is set to 1e-4. An EMA rate of 0.9999 is adopted to stabilize the training. The default parameters are $\alpha = 1.0$, $\beta = 0.2$, and $\gamma = 0.02$ in equation 2, which are determined on a validation set through greedy search. (See Appendix B for more details.) And we set $\eta = 0.2$ in equation 5, $\xi = 0.5$ in equation 6 empirically. During generation, the DDIM sampler is adopted and the default total steps are set to 250. And we can produce a batch (N=128) of sketches by simply calculating the average values of $\varsigma(x_t)$ and $x_{rs}(x_t, s)$ within the batch, hence updating the loss $L_t(s)$ into a batch version accordingly. Using a shared scale within a batch can dramatically reduce the computation cost and speed up the sampling. Additionally, we use an asymmetric reverse process (Kwon et al., 2023) to improve the controllability of classifier guidance, i.e., compute $x_{t-1}$ using the predicted noise before and after performing the guidance together.

## 5.2 RESULTS

**Quantitative Results.** As shown in Table 1, our model outperforms other competitors on all metrics except precision (ours achieves the second best). Interestingly, pixel-based generation methods (i.e., StyleGAN2, DDIM and ours) can clearly beat vector-based approaches. Additionally, our

---

[1]fish, umbrella, apple, moon, shoe, cloud, candle, vase, chair, sun, cat, airplane, spider, car, pig, bus, face, yoga, butterfly, mosquito, lion, television, basket, barn, angel, pizza, book, grapes, fireplace, cake

[2]The guidance scale for DDIM is set to 0.4 determined by greedy search, offering its best FID score.

[3]The optimal guidance strength $\omega$ of CFDG is set to 2 for sketch generation.

Table 1: Quantitative comparison results on *QuickDraw* Dataset. The best and second best are color-coded in red and blue, respectively.

| Model | Random 30 Categories | | | | | 345 Categories | | |
|---|---|---|---|---|---|---|---|---|
| | FID↓ | CLIP-Score↑ | CLIP-Fine(%)↑ | Prec↑ | Rec↑ | FID↓ | Prec↑ | Rec↑ |
| SketchRNN | 8.15 | 0.59 | 52.67 | 0.37 | 0.22 | 10.32 | 0.26 | 0.24 |
| SketchHealer | 5.85 | 0.63 | 51.51 | 0.67 | 0.12 | – | – | – |
| SketchAA | 5.98 | 0.59 | 50.41 | 0.51 | 0.17 | – | – | – |
| SketchKnitter | 7.05 | 0.55 | 43.15 | 0.41 | 0.19 | – | – | – |
| ChiroDiff | 4.75 | 0.59 | 53.16 | 0.64 | 0.18 | 3.17 | 0.58 | 0.25 |
| StyleGAN2 | 4.12 | 0.67 | 53.39 | 0.55 | 0.24 | 2.93 | 0.63 | 0.27 |
| DDIM | 4.08 | 0.67 | 54.19 | 0.71 | 0.30 | 2.85 | 0.74 | 0.31 |
| CFDG | 3.75 | 0.68 | 54.86 | 0.68 | 0.32 | 2.64 | 0.73 | 0.36 |
| Ours | 3.08 | 0.68 | 55.54 | 0.68 | 0.35 | 2.51 | 0.72 | 0.39 |

model achieves the highest CLIP-Score, indicating that the produced sketches by our model can best align the visual content of real sketches. Notably, a higher CLIP-Fine score (i.e., 55.5%) implies that our model tends to produce richer visual content. Some examples of the generated sketches and the retrieved captions are shown in Figure 14, which showcase how our generated sketches can align with the captions summarized from the real sketches.

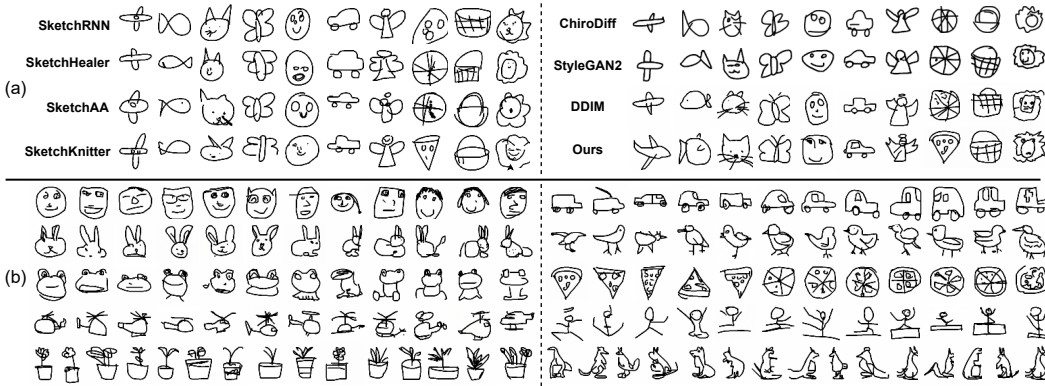

Figure 3: (a) Qualitative comparison results. (b) More generation results by our model.

**Qualitative Results.** Some qualitative results are shown in Figure 3. From Figure 3(a), we can observe that: (i) sketches generated by our model are of better quality in terms of expressiveness, see the drawn `whiskers` of `cat`. (ii) Our method is also capable of depicting objects with more details, see the drawn `antennae` of a `butterfly` and the `window` of a `car` by our model, while these subtle parts are absent from the sketches obtained by other baseline methods. (iii) the sketches produced by our method are more visually appealing and recognizable, e.g., the `eyes` and `nose` on the `human face` are more vividly portrayed, and the overall visual appearance of `lion` is better and more identifiable. More samples generated by our approach can be found in Figure 3(b) and Appendix F. We also visualize an example of the sketch generation process in Figure 4 to better understand the effects of different sampling phases. We can observe that the overall shape of the expected sketch is formed during the warm-up sampling. The scale adaptive guidance sampling instantiates the generation according to the classifier guidance, yielding a sketch of a desired class that fits the overall shape formed in the previous stage. The last phase (i.e., end-up denoising) is responsible for further refinement.

## 5.3 ABLATION STUDY

**Computation analysis and sampling acceleration.** The number of denoising steps during generation is squeezed to 250 using the linear selection procedure following DDIM. To testify how the selection procedure and total sampling steps trade off the overall generation quality and the computational cost, we compare the results under different settings. Results in Table 2a reveal that the linear procedure is typically better than its quadratic counterpart. More sampling steps lead to improved FID and precision but with slightly reduced recall. However, the computational cost is obviously

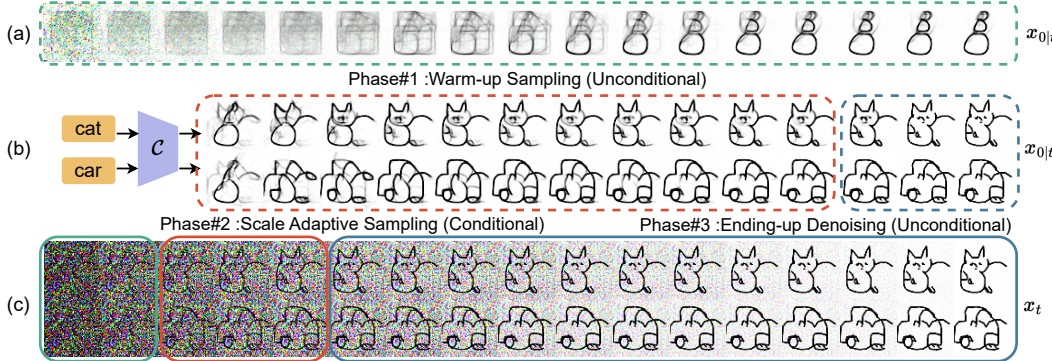

Figure 4: Visualization of $x_{0|t}$ and $x_t$. (a) The estimated final obtained sketches $x_{0|t}$ at time step $t$ during warm-up sampling (green box). The overall structure has been formed in this phase. (b) Given different class labels, i.e., cat and car, $x_{0|t}$ is gradually transformed into the corresponding sketch object by scale adaptive sampling (red box). The end-up denoising (blue) can further refine the sketches by removing the blur in the background. (c) Sketches generated at different time steps.

Table 2: Ablative studies on applying different (a) skip procedures and (b) sampling phases.

(a) Computation analysis. Acceleration in gray.

| Procedure | Steps | FID↓ | Prec↑ | Rec↑ | Speed (s)↓ |
|---|---|---|---|---|---|
| Quadratic | 100 | 8.85 | 0.58 | 0.48 | 0.86 |
| Linear | 100 | 4.95 | 0.62 | 0.48 | 0.90 |
| Quadratic | 250 | 5.54 | 0.61 | 0.32 | 1.87 |
| Linear | 250 | 3.08 | 0.68 | 0.35 | 1.90 |
| Shortcut | 67 | 3.30 | 0.67 | 0.36 | 0.98 |

(b) Effect of each sampling phase.

| Model Variant | FID↓ | CLIP-Score↑ | Prec↑ | Rec↑ | Speed (s)↓ |
|---|---|---|---|---|---|
| **No warm-up** | 3.22 | 0.63 | 0.69 | 0.32 | 2.87 |
| **No Adaptive** | 3.54 | 0.62 | 0.67 | 0.42 | 3.07 |
| **Full Guidance** | 4.08 | 0.67 | 0.71 | 0.30 | 3.27 |
| **No End-up Denoising** | 3.24 | 0.62 | 0.68 | 0.25 | 5.74 |
| **Our Full Model** | 3.08 | 0.68 | 0.68 | 0.35 | 1.90 |

increased in this case. To accelerate the sampling, we will later show that the end-up denoising can occupy up to about $86\%$ of the total sampling steps that can be dramatically shortened.

**Effect of each sampling phase.** There are three phases in order during sampling, i.e., the warm-up sampling, the scale adaptive sampling, and the ending-up denoising sampling. To verify the effectiveness of each phase, we evaluate the generation results in different scenarios. Specifically, (i) *No Warm-up:* Without performing the warm-up sampling, we directly perform scale adaptive sampling at the beginning, followed by the end-up unconditional denoising till the end; (ii) *No Adaptive:* We keep all sampling phases unchanged (i.e., the start and end of each sampling stage remain unchanged) but switch the scale adaptive sampling to the vanilla classifier-guided sampling, i.e., applying a constant gradient scale; (iii) *Full Guidance:* All generation steps are classifier-guided samplings with a constant gradient scale. (iv) *No End-up Denoising:* Sketch generation starts with the warm-up sampling, followed by scale adaptive classifier-guided sampling till the end (i.e., 250 steps are reached).

The results are shown in Table 2b. Compared with our full model, we can find out that (i) *No Warm-up:* Both the FID and recall are getting worse when without the warm-up sampling, indicating that carrying out unconditional generation at the beginning can benefit both fidelity and diversity; (ii) *No Adaptive:* Applying a constant scale (i.e., $s = 0.4$) to the classifier gradients will clearly harm the quality of generation, i.e., fidelity (FID). (iii) *Full Guidance:* Merely performing classifier guidance with a constant gradient scale (i.e., $s = 0.4$) will simultaneously lower the fidelity and diversity; (iv) *No End-up Denoising:* Both the fidelity and mode coverage of the produced sketches are compromised when maintaining the classifier guidance till the end. This is because too strong classifier guidance can lead to over-sketching, hence resulting in declined sample quality and diversity. Moreover, the generation is much more expensive (i.e., 5.74 s per sketch) in this case due to the increased sampling steps required for gradient scale optimization.

**Length of each sampling phase.** To study the influence of varying the length of each sampling phase, we compare the generation results using different configurations of the parameters $\eta$ and $\xi$ in Eq. (5) and Eq. (6). Results are shown in Table 3. We can find out that most steps are occupied by the end-up denoising sampling, which can be shortened for acceleration as shown in the last row in Table 3. When increasing $\xi$ and fixing $\eta$, the length of the scale adaptive sampling becomes longer,

Table 3: Comparison results when varying the parameters $\eta$ and $\xi$.

| # Steps | $\eta$ | $\xi$ | Warm-up (%) | Adaptive(%) | End-up(%) | FID↓ | Prec↑ | Rec↑ | Speed (s)↓ |
|---|---|---|---|---|---|---|---|---|---|
| | | 0.3 | 6.01 | 8.26 | 85.73 | 3.21 | 0.65 | 0.38 | 1.76 |
| | 0.2 | 0.4 | 6.01 | 10.02 | 83.97 | 3.17 | 0.66 | 0.37 | 1.83 |
| | | 0.5 | 6.01 | 11.73 | 82.25 | 3.08 | 0.68 | 0.35 | 1.90 |
| T=250 | 0.1 | | 4.47 | 12.56 | 82.97 | 3.12 | 0.69 | 0.32 | 2.03 |
| | 0.3 | 0.5 | 13.27 | 7.52 | 79.21 | 5.37 | 0.61 | 0.42 | 1.78 |
| | 0.4 | | 18.52 | 5.44 | 76.04 | 8.53 | 0.57 | 0.48 | 1.67 |
| T=67 (shortcut) | 0.2 | 0.5 | 15.22 | 40.90 | 43.88 | 3.30 | 0.67 | 0.36 | 0.98 |

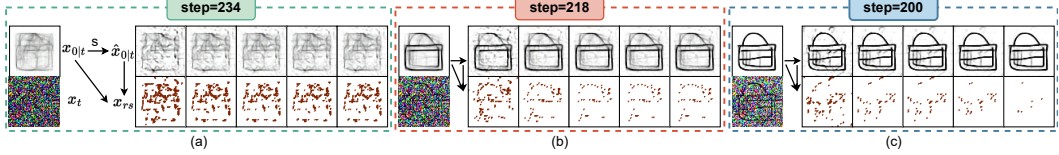

Figure 5: Visualization of the residual sketches $x_{rs}$ and the estimated final sketch $\hat{x}_{0|t}$ during scale optimization at (a) early (b) middle and (c) late sampling steps. By scale optimization, the residual sketches are getting more organized and cleaner.

leading to improved sample quality (i.e., lower FID score) yet narrowing the mode coverage (i.e., reduced recall). Extending the warm-up sampling will squeeze the scale adaptive sampling, and let the sample quality get worse but achieve better diversity. The optimal balance is reached when warm-up takes about half the number of the adaptive sampling steps.

**Visualization of scale optimization.** To better understand the mechanism of utilizing scaling indicator as an explicit signal to obtain the optimized gradient scale, we visualize the optimization process along with the corresponding residual sketches $x_{rs}(x_t, s)$ and the estimated final sketch $\hat{x}_{0|t}$. As shown in Figure 5, at the beginning of optimization, the randomly initiated guidance scale $s$ is often mismatched with the scaling indicator $\varsigma(x_t)$. The corresponding residual sketch $x_{rs}(x_t, s)$ looks messy and unstructured, implying a less favorable (i.e., too noisy) output sketch $\hat{x}_{0|t}$ under the classifier guidance. Once the gap between $\varsigma(x_t)$ and $x_{rs}(x_t, s)$ is closed, the residual sketch becomes cleaner and more organized. As a result, the expected sketches $\hat{x}_{0|t}$ using the optimized scale are painted in a more structured and concise way.

## 6 CONCLUSION

Raster sketches generated by diffusion models using classifier guidance with a constant scale are sub-optimal, either too sparse to recognize or too densely depicted (i.e., over-sketching). We show that the generation quality can be improved by simply adjusting the guidance scale dynamically at each sampling step, without retraining the model. Concretely, we proposed to optimize the scale according to the predictable generation results at each sampling step by using the developed scale indicator and residual sketch. It is observed that the pixel changes, i.e., the residual sketches, during sampling are more organized and located at critical positions to form a sketch object by our model. Injecting unconditional sampling at the beginning and the end of generation is also beneficial. Uniquely, we proposed to assess the generated sketches in terms of expressiveness by using the CLIP-Score. It shows that ours can generate sketches containing richer object details.

ACKNOWLEDGMENTS

This work was supported by NSFC under No.61601042 and the Program for Youth Innovative Research Team of BUPT under No. 2023QNTD02.

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

## A    DDIM SAMPLING

DDIM (Song et al., 2020a) reveals that the learning objective of DDPM (Ho et al., 2020) only depends on the "marginals" $q(x_t|x_0)$ rather than the joint $q(x_{1:T}|x_0)$, thus a non-markovian forward process is proposed such that the desired marginals are fulfilled, yielding the same learning objective as DDPM. Formally, the forward process of DDIM is formulated as:

$$q(x_t|x_{t-1}, x_0) = \frac{q(x_{t-1}|x_t, x_0)q(x_t|x_0)}{q(x_{t-1}|x_0)} \tag{7}$$

where $q(x_t|x_0)$ and $q(x_{t-1}|x_0)$ can be obtained by the marginals:

$$q(x_t|x_0) = \mathcal{N}(x_t; \sqrt{\bar{\alpha}_t}x_0, (1 - \bar{\alpha}_t)\mathbf{I}) \tag{8}$$

and $q(x_{t-1}|x_t, x_0)$ is a Gaussian defined in the following:

$$q(x_{t-1}|x_t, x_0) = \mathcal{N}(x_{t-1}; \underbrace{\sqrt{\bar{\alpha}_{t-1}}x_0 + \sqrt{1 - \bar{\alpha}_{t-1} - \sigma_t^2} \cdot \frac{x_t - \sqrt{\bar{\alpha}_t}x_0}{\sqrt{1 - \bar{\alpha}_t}}}_{Mean}, \sigma_t^2\mathbf{I}) \tag{9}$$

where the choice of mean function is to ensure the desired marginals, i.e., $q(x_t|x_0) = \mathcal{N}(x_t; \sqrt{\bar{\alpha}_t}x_0, (1-\bar{\alpha}_t)\mathbf{I})$ for all $t$. Then during the generative process, $q(x_{t-1}|x_t, x_0)$ can be used to approximate the denoising function $p_\theta(x_{t-1}|x_t)$ without knowing $x_0$, which can be predicted from $x_t$ at timestep $t$ derived from Eq. (8):

$$\hat{\boldsymbol{x}}_0 = (\boldsymbol{x}_t - \sqrt{1 - \alpha_t} \cdot \epsilon_\theta^{(t)}(\boldsymbol{x}_t))/\sqrt{\alpha_t} \tag{10}$$

Consequently, derived from Eq. 9, we can sample data by repeating:

$$x_{t-1} = \sqrt{\bar{\alpha}_{t-1}} \left( \frac{x_t - \sqrt{1 - \bar{\alpha}_t}\epsilon_\theta(x_t, t)}{\sqrt{\bar{\alpha}_t}} \right) + \sqrt{1 - \bar{\alpha}_{t-1} - \sigma_t^2} \cdot \epsilon_\theta(x_t, t) + \sigma_t\epsilon_t \tag{11}$$

DDIM shows that sampling could be accelerated using much fewer steps when setting $\sigma_t = 0$ for all timesteps $t$.

## B    HYPERPARAMETER SETTING IN EQUATION 2

The hyperparameters (i.e., $\alpha$, $\beta$, and $\gamma$) of Equation 2 for calculating the scaling indicator are determined by greedy search on a small validation set, comprising of sketches generated by our model. Specifically, we generate 1k sketches per class by conducting classifier guidance on the obtained unconditional DDPM under different hyperparameter configurations. As shown in Figure 6, the optimal values are $\alpha = 1.0$, $\beta = 0.2$, and $\gamma = 0.02$.

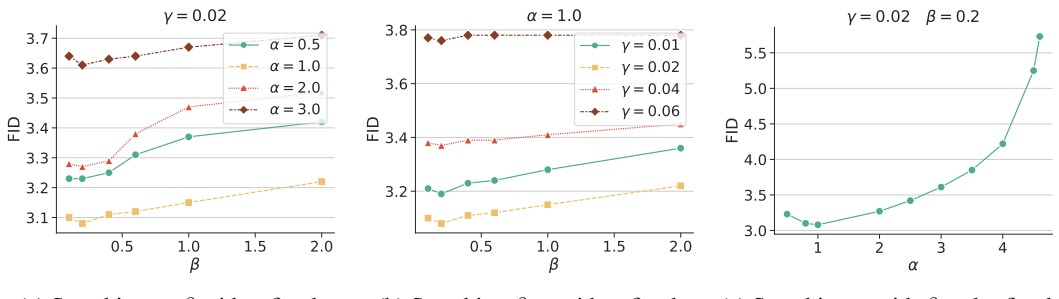

(a) Searching $\alpha, \beta$ with $\gamma$ fixed.    (b) Searching $\beta, \gamma$ with $\alpha$ fixed.    (c) Searching $\alpha$ with $\beta$ and $\gamma$ fixed.

Figure 6: Greedy search on hyperparameters $\alpha$, $\beta$, and $\gamma$ based on FID scores.

## C    LOSS CURVE OF $L_t(s)$ DURING SCALE OPTIMIZATION

To gain more insights into the process of scale optimization, we plot the scaling indicator $\varsigma(x_t^{(i)})$, the fraction of per-pixel difference of the residual sketch, i.e., the $x_{rs}$ term $\frac{1}{HW}\sum_{HW} x_{rs}(x_t^{(i)}, s)$, and their resulting loss $L_t(s)$ at six different time steps which are evenly selected during the stage of scale adaptive sampling. As shown in Figure 7, we can see that $L_t(s)$ can converge quickly after a few SGD optimization steps for various sampling time steps $t$.

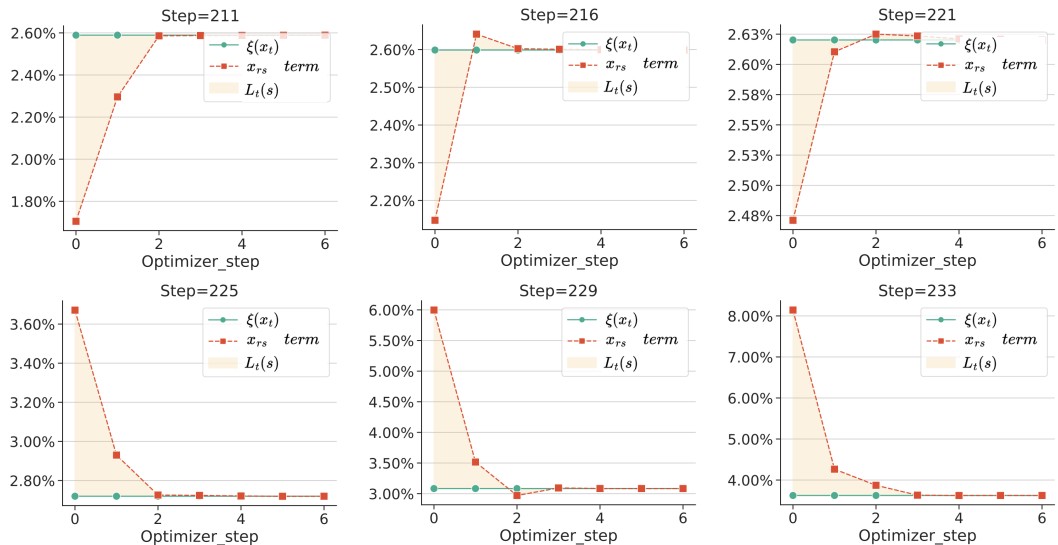

Figure 7: Scaling indicator (denoted by the green line) and the $x_{rs}$ term (denoted by the red dashed line), and their corresponding loss $L_t(s)$ (the gap between the red and green lines) at different steps during scale optimization.

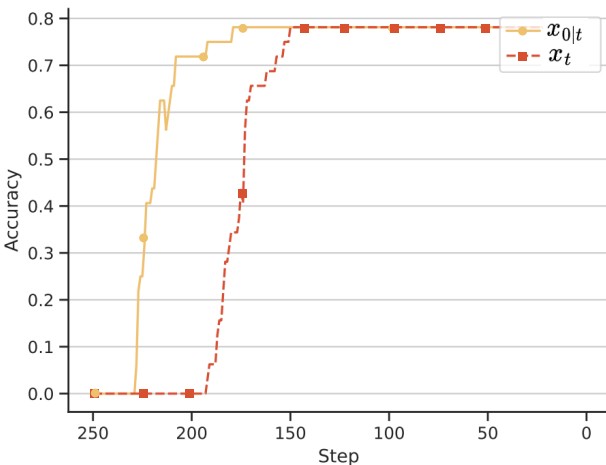

Figure 8: Top-1 classification accuracy on noisy sketches $x_t$ and the estimated sketches $x_{0|t}$.

## D   THE EFFECTIVENESS OF CLASSIFICATION ON $x_{0|t}$

To validate the applicability of the classifier (trained using the noisy sketches $x_t$) to recognize the estimated sketch $x_{0|t}$, we provide the results of the top-1 classification accuracy both on $x_t$ and $x_{0|t}$ using the same classifier. As shown in Figure 8, the estimated sketch $x_{0|t}$ can be successfully recognized by using the classifier, i.e., achieving the same level top-1 accuracy of $x_t$.

# E   SUMMARIZED CAPTIONS

| Category | Prompt | Category | Prompt | Category | Prompt |
|---|---|---|---|---|---|
| airplane | an airplane with two wings | angel | an angel with a pair of wings | apple | a round apple |
| | an airplane with a streamlined fuselage | | an angel with a halo above his head | | an apple with a stem |
| | an airplane flying upward | | an angel in robes | | an apple with one leaf |
| | an airplane with tail fin | | an angel with feather-like wings | | an apple with a small pit on the bottom |
| | an airplane with windows | | an angel with eyes | | an apple with two leaves |
| barn | a barn with a peaked roof | basket | a basket with a handle | book | an open book |
| | a barn with a door | | a plaid woven basket | | a closed book |
| | a barn with double doors | | a square basket | | a book with wavy lines |
| | a barn with arched door | | a round-bottomed basket | | a book with a spine |
| | a barn with windows | | a basket with a thin handle | | a book with a pattern |
| bus | a bus with a rectangular body | butterfly | a butterfly with a pair of wings | cake | a cylindrical cake |
| | a bus with a row of windows | | a butterfly with an elongated body | | a cake with candles |
| | a bus with a row of tires | | a butterfly with antennae | | a cake with two tiers |
| | a bus with doors | | a butterfly with horizontal lines on its body | | a cake with wavy lines on the sides |
| | a bus with an antenna on the roof | | a butterfly with spots on its wings | | a cake with decorations on it |
| candle | a rectangular candle | car | a car with two tires | cat | a round-faced cat |
| | a candle with flame | | a car with a flat roof | | a cat with eyes |
| | a candle with wax drops on the bottom | | a car with a rectangular body | | a cat with pointed ears |
| | a candle with wax drops on the side | | a car with doors | | a cat with beard |
| | a candle with patterned sides | | a car with windows | | a full body cat |
| chair | a four-legged chair | cloud | a fluffy cloud | face | a round face |
| | a chair with an upright back | | a cloud with wavy edges | | a smiling face |
| | a chair with an oval seat | | a cloud with a flat bottom | | a face wearing glasses |
| | a chair with a rectangular seat | | a flat cloud | | a boy's face with short hair |
| | a chair with rails on the back | | a marshmallow-like cloud | | a girl's face with two pigtails |
| fireplace | a rectangular fireplace | fish | a fish with an oval body | sun | a sun with a large circle in the center |
| | a fireplace with a pile of firewood | | a fish with a triangular tail | | a sun with dashes around it |
| | a fireplace with flames | | a fish with eyes | | a sun with wavy lines on the border |
| | an arched fireplace | | a fish with a mouth | | a sun surrounded by triangular rays |
| | a fireplace with chimney | | a fish with fins | | a sun with a smiling face |
| spider | a spider with many legs | television | a rectangular television | grapes | a bunch of round grapes |
| | a spider with an oval body | | a television with an antenna on top | | a bunch of closely packed grapes |
| | a spider with a segmented body | | a television with a base | | a bunch of grapes with leaves |
| | a spider with black body | | a television with a row of buttons | | a bunch of grapes with glucose |
| | a spider with bent legs | | a television with patterns on the screen | | a bunch of grapes of the same size |
| lion | a lion's face | pizza | a round pizza | shoe | a flat shoe |
| | a lion with a thick mane | | a triangular pizza | | a round-toed shoe |
| | a lion with eyes | | a pizza with outer edge | | a boot |
| | a lion with ears | | a pizza with round garnish | | a high heel shoe |
| | a full-body lion | | a pizza with cutting lines | | a shoe with laces |
| vase | a plump vase | yoga | a stickman | moon | a round moon |
| | a cylindrical vase | | a person on a yoga mat | | a small crescent moon |
| | a vase with a flat bottom | | a person with arms raised | | a waxing moon |
| | a vase with an elongated neck | | a person standing on one foot | | a waning moon |
| | a vase with flowers | | a person lying on the ground | | a half-circle moon |
| umbrella | a domed umbrella | pig | a round-faced pig | mosquito | a mosquito with spikes |
| | an umbrella with a curved handle | | a pig with eyes | | a mosquito with wings |
| | an umbrella with ribs | | a pig with an oval nose | | a mosquito with slender legs |
| | an umbrella with wavy edges | | a pig with ears | | a mosquito with an elongated body |
| | an umbrella with horizontal edges | | a full-body pig | | a mosquito with longitudinal texture |

Figure 9: All captions summarized from the randomly chosen 30 categories. Five captions are collected per category with template "*this is a sketch of X*" in our experiments, where X is either a coarse or fine-grained text description, as shown in the table. The fine-grained and coarse captions are color-coded in red and blue, respectively.

## F  MORE SAMPLES

Figure 10: Samples from our model. Classes are yoga, kangaroo, house plant.

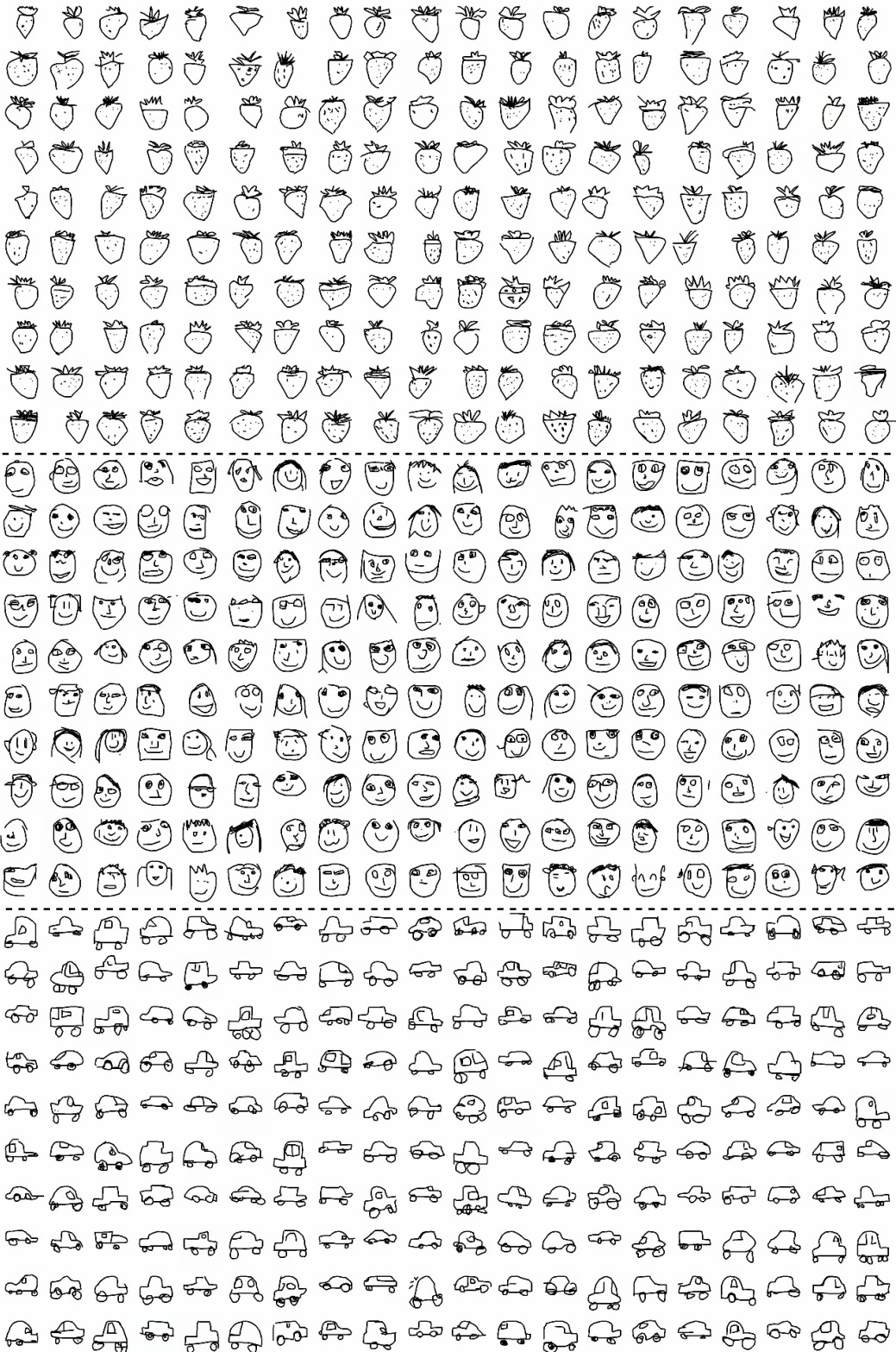

Figure 11: Samples from our model. Classes are strawberry, face, car.

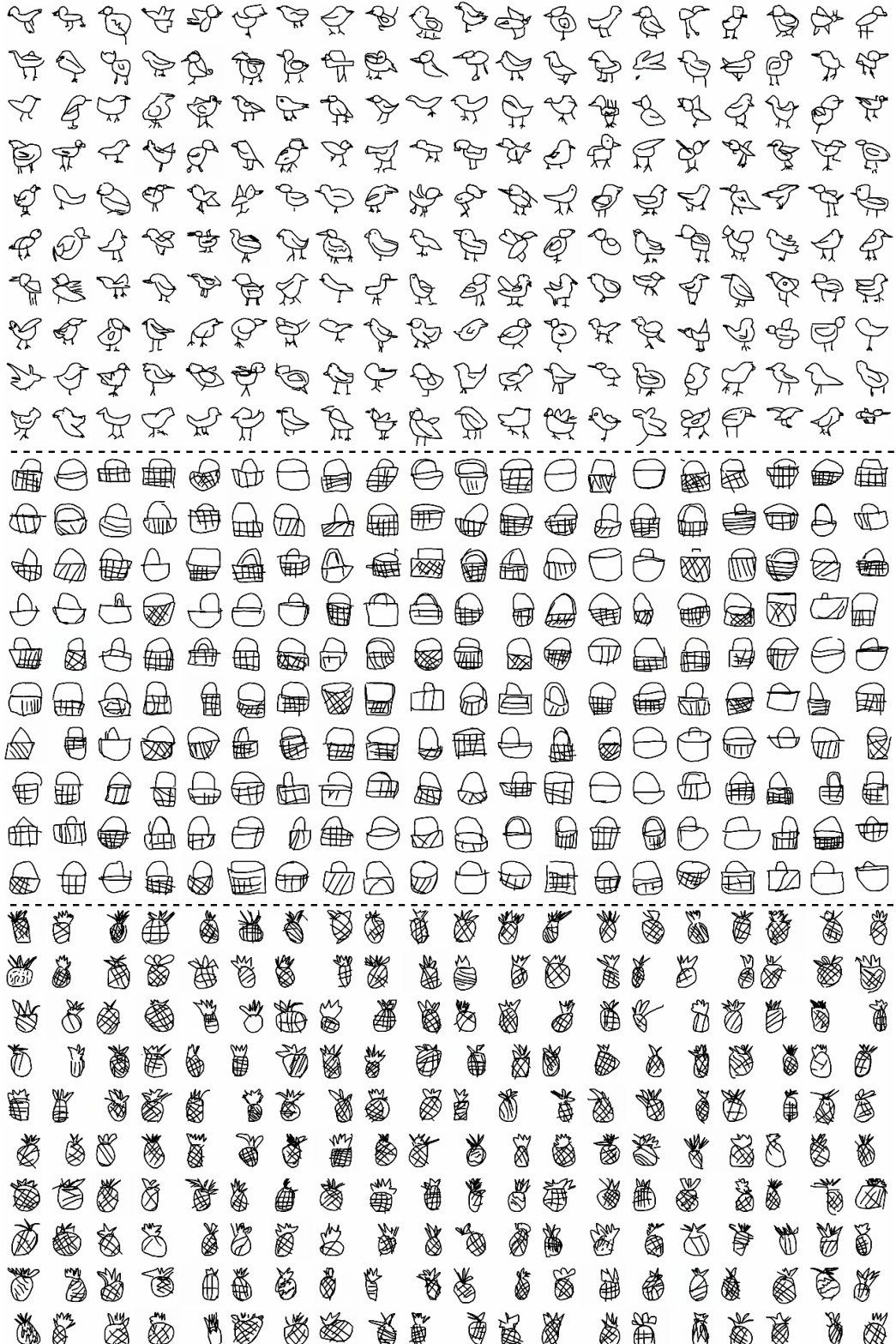

Figure 12: Samples from our model. Classes are bird, basket, pineapple.

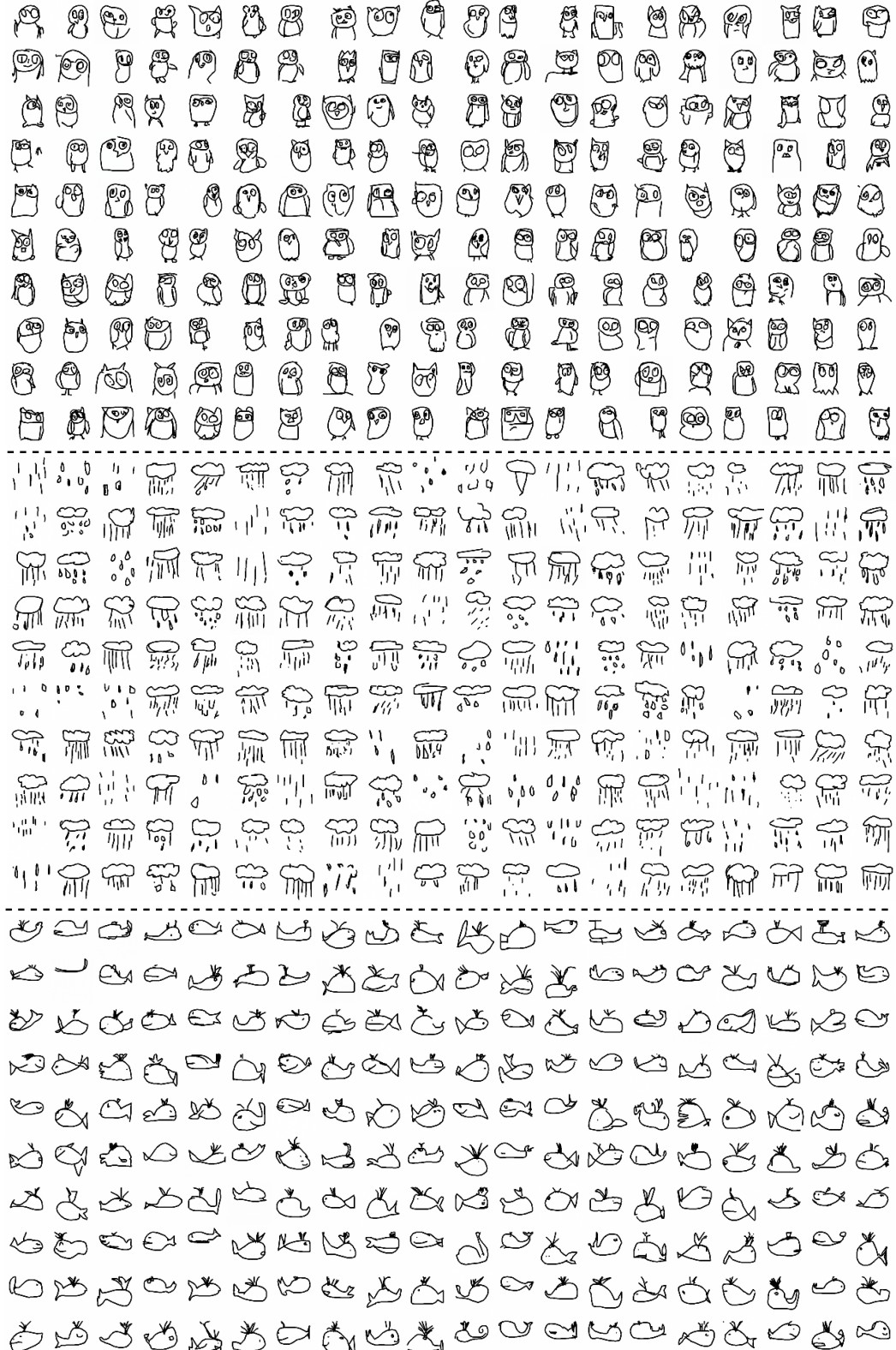

Figure 13: Samples from our model. Classes are owl, rain, whale.

## G  EFFECT OF BATCH SIZE

To inspect the robustness of our model under various choices of batchsize, and how the batchsize impact the results, we set different batchsize to generate 5k samples for evaluation. As shown in the table 4, the FID score remains relatively stable while the recall increases clearly when the batchsize becomes smaller (but much more time-consuming). The trend of generation results can conform to the expectation, i.e., using smaller batchsize offers improved results.

When setting batchsize=1, other than the known fact that this choice can highly slow down the sampling process , we further found that it makes the sampling sometimes unstable due to gradient vanishing during scale optimization, i.e., the scale gradient becomes zero (achieving local minimum). In contrast, our default setting, i.e., batchsize=128, offers alleviation to the sisue since larger batches provide more samples for gradient computation thus slowing down the vanishing of gradients.

To sum up, batchsize=1 should be ideal for scale optimization theoretically, however it is suboptimal due to the issue of scale gradient vanishing in practice. As a result, we altered to a batch version scale for speeding up and stabilizing the generation.

Table 4: Comparison results when varying batchsize on 30 categories of *QuickDraw* datasets.

| batchsize | FID↓ | Prec↑ | Rec↑ | Speed(s)↓ |
|---|---|---|---|---|
| 1 | 3.34 | 0.67 | 0.42 | 274.57 |
| 16 | 3.13 | 0.68 | 0.40 | 17.62 |
| 64 | 3.11 | 0.68 | 0.38 | 3.97 |
| 128 | 3.08 | 0.68 | 0.35 | 1.90 |

## H  EXAMPLES AND THE RETRIEVED CAPTIONS USING CLIP-SCORE

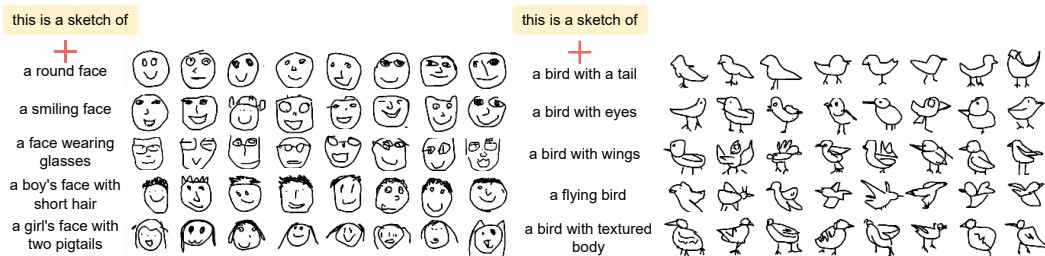

Figure 14: Examples of the generated sketches and the retrieved captions (Top-1) using CLIP-Score.

