# OpenReview forum: "Scale-Adaptive Diffusion Model for Complex Sketch Synthesis"
_ICLR.cc/2024/Conference — ICLR 2024 poster_

### Official Review · Reviewer_SPRZ · 2023-10-29

**Soundness:** 3 good
**Presentation:** 3 good
**Contribution:** 2 fair
**Rating:** 6
**Confidence:** 3

**Summary:**

The authors proposed a strategy to dynamically change the scale of classifier-guidance during the reverse diffusion process for the generation of pixel-based sketches. For this purpose, they empirically designed a scaling indicator and a residual sketch to achieve the optimal guidance scale needed at the current diffusion stage. The scaling indicator is to assess the recognizability and complexity of current generative results. The residual sketch x_rs (x_t, s) is to evaluate the extent to which the guidance scale s will impact the generative process. The guidance scale s is optimized to enforce the residual sketch to be synchronized with the scale indicator. Moreover, they adopted a three-phase sampling strategy to enhance the sketch diversity and quality. Experiments demonstrated the superiority of their approach in the realm of sketch generation.

**Strengths:**

1. The proposal of scaling indicator and residual sketch is novel and can conduct an effective scale adaptive classifier-guided diffusion process.
2. The experimental part is convincing, and includes both quantitative and qualitative results. And the ablation study shows the effectiveness of their methods.
3. The organization of this paper is clear and easy to understand.

**Weaknesses:**

1. The scale adaptive classifier-guided diffusion process seems useful and attractive, however, the scaling indicator and residual sketch are proposed mainly based on their experience in the field of sketch generation. In my opinion, it may rely on the simple structure of sketch. I wonder if similar methods can be applied to other data modalities, such as natural images.
2. Equation 4 appears to be dubious. I understand the point that the residual sketch should be synchronized with the scale indicator. However, are they on the same numerical scale and is it appropriate to directly apply mean squared error for optimization? I believe that a more rigorous discussion or proof is needed based on their definitions (Formula 2 and 3).
3. The contribution “three-phase sampling strategy” has been proven to be effective but it is merely a simple technique for diffusion process and the innovation may be limited.
4. Using the fraction of stroke pixels to the whole canvas to evaluate the sketch complexity is oversimplified.

**Questions:**

1. The authors mentioned that vector-base approaches are inherently limited when tackling intricate and complex sketches. I would like to understand why pixel-based methods have an advantage over vector-based methods? I hope the authors can provide some intuitive explanations.
2. In Section 2, the authors claimed that an additional property of pixel-based diffusion modeling is classifier gradient can be used as guidance. So are there any obstacles to using classifier-guidance for vector-based diffusion? It also seems feasible in principle.
3. In Equ 2, are the settings of the three hyperparameters obtained through experiments or are there some empirical principles?
4. In Equ. 4, why do different images in the same batch share the same scale?

---

> ### Author Response · Authors · 2023-11-22
> **Response to Reviewer SPRZ**
>
> Thank you for your time and critical feedback!
> >The scale adaptive classifier-guided diffusion process seems useful and attractive, however, the scaling indicator and residual sketch are proposed mainly based on their experience in the field of sketch generation. In my opinion, it may rely on the simple structure of sketch. I wonder if similar methods can be applied to other data modalities, such as natural images.
>
> Interesting point! Unfortunately, the model proposed in this work is sketch-specific as one of the key components, i.e., complexity, is measured by the fraction of the *binary* stroke pixels on the whole canvas. Such an assumption is no longer held for natural images. However, perhaps the take-home message is that imposing some explicit quality control over the generation is possible by leveraging scale adaptive classifier guidance.
>
> >Equation 4 appears to be dubious. I understand the point that the residual sketch should be synchronized with the scale indicator. However, are they on the same numerical scale and is it appropriate to directly apply mean squared error for optimization? I believe that a more rigorous discussion or proof is needed based on their definitions (Formula 2 and 3).
>
> Thanks for the suggestion. We have now provided the scores of these two terms in Equation 4 at different time steps during optimization in Appendix C in the revised paper. It shows that the numerical scales are on the same level, which is achievable because the parameters (i.e., $\alpha$, $\beta$ and $\gamma$) can adapt the scale indicator $\varsigma(x_t^{(i)})$ to be on par with the residual sketch term, i.e., ${1\over HW}\sum_{HW} x_{rs}(x_t^{(i)}, s)$.
>
> > The contribution “three-phase sampling strategy” has been proven to be effective but it is merely a simple technique for diffusion process and the innovation may be limited.
>
> Thanks. Yes, simple as it might be, this does align with our vision of coming up with a more explainable approach, and we hope you would agree that, the results are quite decent (i.e., it does work).
>
> > Using the fraction of stroke pixels to the whole canvas to evaluate the sketch complexity is oversimplified.
>
> Thanks, we were also surprised at how effective this simple strategy is – this is however sketch-specific (black and white pixels), and chance is it will not generalize to natural images.
>
> >The authors mentioned that vector-base approaches are inherently limited when tackling intricate and complex sketches. I would like to understand why pixel-based methods have an advantage over vector-based methods? I hope the authors can provide some intuitive explanations.
>
> Of course! Intuitively, we seek to sidestep the cost-expensive process of rasterization, i.e., neural line rendering [A], which is a common practice for existing works on vector-based sketch generation [A, B]. Rasterization is often required since the classifier is trained on pixel-format sketches. We will further clarify it in the revised version.
>
> [A] "Sketch-R2CNN: an RNN-rasterization-CNN architecture for vector sketch recognition." TVCG 2020.
> [B] "SketchKnitter: Vectorized Sketch Generation with Diffusion Models." ICLR 2023.
>
> > In Section 2, the authors claimed that an additional property of pixel-based diffusion modeling is classifier gradient can be used as guidance. So are there any obstacles to using classifier-guidance for vector-based diffusion? It also seems feasible in principle.
>
> Thanks. The trouble here is that the classifier is implemented by CNN, which is obtained by training on (rasterized) sketch images. Hence, the generated sketches (in vector format) will have to be converted back to rasterized images before applying the classifier-guidance, which is costly. This process can be even more expensive since it demands multiple classifier-guided sampling during our scale optimization at each time step.
>
> > In Equ 2, are the settings of the three hyperparameters obtained through experiments or are there some empirical principles?
>
> We perform a greedy search strategy to determine the optimal choice of $\alpha$, $\beta$ and $\gamma$. We have added more details in the implementation and further provided ablative results when varying these hyperparameters in Appendix B.
>
> > In Equ. 4, why do different images in the same batch share the same scale?
>
> Thanks! In our implementation, a batch version scale is utilized in order to reduce the computational cost, due to limited GPU resources. Ideally, each sketch generation should have its own scale schedule. However, this can be readily achieved by setting the batch size to one, but at a higher cost as noted. We have provided more discussions on this in the implementation details of the paper revision.

---

### Official Review · Reviewer_t4As · 2023-10-30

**Soundness:** 3 good
**Presentation:** 3 good
**Contribution:** 3 good
**Rating:** 6
**Confidence:** 4

**Summary:**

This paper proposes a novel sketch generation method based on classifier-guided diffusion models. Specifically, the authors propose a scale-adaptive classifier-guided diffusion model, which achieves a delicate balance between recognizability and complexity in generated sketches. In addition, the authors also propose a three-phase sampling strategy to enhance sketch diversity and quality.

**Strengths:**

+ The authors analyze the impact of guidance scale on diffusion-based sketch generation tasks and propose a scale adaptive classifier-guided sampling method to achieve a delicate balance between recognizability and complexity in generated sketches.
+ The authors point out the impact of unconditional guidance and classifier guidance on generating sketches in diffusion models and propose a three-phase sampling strategy.
+ Quantitative and qualitative experiments have shown that the proposed method outperforms existing sketch generation methods.

**Weaknesses:**

+ From Figure 2 and Section 4.1, we can see that the input of the sketch classifier is a clean sketch estimated from noisy images. However, the author mentioned in Section 4.1 that the classifier is trained by using noisy sketches, which is obviously contradictory.

+ From Table 2, we can see that using unconditional guidance can increase the diversity of generated sketches. However, from Figure 5, it can be seen that under the same random seeds, the images generated by different categories during the warm-up sampling stage are the same, and some strokes will remain in the final generation result. Therefore, it remains to be discussed whether the warm-up sampling is truly effective for sketch generation tasks.

**Questions:**

The default size of the produced sketches is set to 64×64. Can this method still generate sketches well at higher resolutions?

---

> ### Comment · Reviewer_t4As · 2023-11-22
> **The comment during rebuttal**
>
> Since I don't receive a timely response from the authors during the rebuttal period, I will maintain my original rating score.

---

> > ### Author Response · Authors · 2023-11-22
> >
> > We apologize for the delayed response due to the additional experiments. Hope our replies and paper revisions have addressed your concerns.

---

> ### Author Response · Authors · 2023-11-22
> **Response to Reviewer t4As**
>
> Thank you for the critical feedback. Below are our replies to each of your questions.
> > From Figure 2 and Section 4.1, we can see that the input of the sketch classifier is a clean sketch estimated from noisy images. However, the author mentioned in Section 4.1 that the classifier is trained by using noisy sketches, which is obviously contradictory.
>
> Thanks! To clarify, the estimated sketch $x_{0|t}$ at each time step $t$ is not a clean one (notice the blur regions of $x_{0|t}$ in Figure 2), especially in the early stage $x_{0|t}$ can be rather noisy.
>
> To gain more insights into this matter, we have provided some quantitative results of using the classifier to recognize $x_{0|t}$ at different time steps in Appendix D. For comparison, we also report the classification results on $x_t$ (i.e., noisy sketches). We can see that the classifier can achieve the same level of top-1 accuracy both on $x_t$ and $x_{0|t}$.
>
>
>
> > From Table 2, we can see that using unconditional guidance can increase the diversity of generated sketches. However, from Figure 5, it can be seen that under the same random seeds, the images generated by different categories during the warm-up sampling stage are the same, and some strokes will remain in the final generation result. Therefore, it remains to be discussed whether the warm-up sampling is truly effective for sketch generation tasks.
>
> Good point! Actually, warm-up sampling can indeed increase diversity of the holistic structure of the final generated sketches, when compared to the alternative of applying classifier guidance at the beginning, as shown in Table 2(b).
>
> In Figure 5, we meant to show that classifier guidance can be applied starting at the second stage, i.e., scale adaptive sampling, where the model can adapt to whatever the overall shape (diversity enlarged) formed at the first warm-up stage.
>
> > The default size of the produced sketches is set to 64×64. Can this method still generate sketches well at higher resolutions?
>
> Sure, we can change the default resolution to a higher one, and we have conducted additional experiments of generating sketches in resolution of $256\times 256$ on the same subset of 30 categories. Results show that the quality of the generated sketch remains at the same level (FID 3.76).

---

### Official Review · Reviewer_bpbS · 2023-10-30

**Soundness:** 3 good
**Presentation:** 3 good
**Contribution:** 2 fair
**Rating:** 6
**Confidence:** 4

**Summary:**

This paper divides the diffusion denoising process into three phases. The first and the last phases are unconditional generation, the middle phase is conditioned on the classifier guidance. For the middle phase, this paper proposes an extension block for classifier-guided diffusion models to perform scale adaptive classifier-guided sampling on the task of pixel-level sketch generation, addressing the challenge of dynamically optimizing the guidance scale for classifier-guided diffusion. This method achieves a balance between "recognizability" and "complexity" in the generated sketches. Experiments are on the QuickDraw dataset.

**Strengths:**

1. The three-phase sampling strategy can maintain sketch diversity and quality.

2. The idea of dynamically optimizing the guidance scale is reasonable and prospective.

3. Overall, the presentation and writing are easy to follow, and the experiment is detailed.

**Weaknesses:**

1. The main weakness is that the proposed block is specialized for classifier-guided diffusion. However, classifier-guided diffusion models are not as popular and powerful as those multimodal diffusion models using cross-attention. This limits the impact and universality of the proposed block.

2. The idea of scale adaptive classifier-guided sampling is sound and this method achieves a balance between "recognizability" and "complexity". However, this also fixes the "recognizability" and "complexity" of the results. I mean, in classifier-guided diffusion models, users can adjust the guidance scale to trade off diversity for fidelity. But the proposed adaptive method can not.

3. The implementation details are naive such as the complexity c(x0|t) using $L_0$ norm, and the determination equotion for $t_w$ and $t_d$.

**Questions:**

1. I wonder whether the SGD process employed to obtain the optimal value of guidance scale s at each timestep t by minimizing Lt(s) is convergent. There is no further exploration.

2. Also about the SGD process. In equation 4,"N is the number of sketches generated within a sampling batch". But why should the sketches in the same batch share the same guidance scale $s$? I mean, $s$ should be independent for each sketch in one batch.

3. About the optimization objective Lt(s), which is "intuitive" but not from mathematical proof. This optimization objective seems questionable.

4. How to you determine the parameters α = 1.0, β = 0.2, and $\gamma$ = 0.02?

---

> ### Author Response · Authors · 2023-11-22
> **Response to Reviewer bpbS**
>
> Thank you for the valuable comments!
> > The main weakness is that the proposed block is specialized for classifier-guided diffusion. However, classifier-guided diffusion models are not as popular and powerful as those multimodal diffusion models using cross-attention. This limits the impact and universality of the proposed block.
>
> Thanks! Indeed, cross-modal diffusion models, such as stable diffusion, are more appealing than classifier-guided models for controllable generation. However, our work enables a fine-tuned control of the sketch generation to achieve a complexity and recognizability balance by adaptive scaling and the residual sketch, which is currently under-explored by existing diffusion models. To explore if such a finer tuning strategy is applicable to cross-modal diffusion models will be our future endeavor.
>
> > The idea of scale adaptive classifier-guided sampling is sound and this method achieves a balance between "recognizability" and "complexity". However, this also fixes the "recognizability" and "complexity" of the results. I mean, in classifier-guided diffusion models, users can adjust the guidance scale to trade off diversity for fidelity. But the proposed adaptive method can not.
>
> Great point! Although the vanilla classifier-guided diffusion model offers users the "flexibility" to trade off diversity for fidelity, this process is cumbersome or even impossible in practice as revealed in Figure 1. Users have to try numerous choices for each different class. And there is no way to identify if the optimal or even nearby optimal is achieved. In contrast, our proposed method automates scale searching. Importantly, the whole process is explainable due to the interpretability of the scale indicator and the visibility of the devised residual sketch, as shown in Figure 6.
>
> > The implementation details are naive such as the complexity $c(x_{0|t})$ using $L_0$ norm, and the determination equotion for $t_w$ and $t_d$.
>
> Thanks, we think this is rather neat, and most importantly matches our vision of enabling a more explainable approach. Results too are fairly convincing, we hope.
>
> > I wonder whether the SGD process employed to obtain the optimal value of guidance scale s at each timestep t by minimizing Lt(s) is convergent. There is no further exploration.
>
> Great suggestion! We have now provided the loss values at different time steps during scale adaptive sampling in Appendix C. We can see that, given a scaling indicator $\varsigma(x_t^{(i)})$ as target, the fraction of pixel changes, i.e., ${1\over HW}\sum_{HW} x_{rs}(x_t^{(i)}, s)$, is forced to approach the target by optimizing the scale $s$, hence the loss $L_{t}(s)$ is gradually converged to zero.
>
>
> > Also about the SGD process. In equation 4,"N is the number of sketches generated within a sampling batch". But why should the sketches in the same batch share the same guidance scale  $s$? I mean,  $s$  should be independent for each sketch in one batch.
>
> Good spot! It is correct that $s$ should ideally be independent for each sketch. However, in order to speed up training and the later generation process (given the limited computation resource), we then sought to optimize $L_t(s)$ in a batch, as we described in the implementation details. Even with this compromised choice, our model can still achieve superior generation results over the baseline methods.
>
> > About the optimization objective Lt(s), which is "intuitive" but not from mathematical proof. This optimization objective seems questionable.
>
> Thanks. This objective function could be interpreted as a regression problem, which synchronizes the residual sketch to the scaling indicator, thus determining the optimal scale $s$. Intuitively, minimizing $L_{t}(s)$ is to enforce the fraction of pixel changes reflected by the residual sketch, i.e., ${1\over HW}\sum_{HW} x_{rs}(x_t^{(i)}, s)$, can conform to the scaling indicator, i.e., $\varsigma(x_t^{(i)})$, which sets a goal of balanced complexity and recognizability for each sampling step.
>
> > How to you determine the parameters $\alpha$ = 1.0, $\beta$ = 0.2, and  $\gamma$  = 0.02?
>
> The optimal parameters are determined by performing greedy search on a small validation set, which is composed of sketches generated by our model. Specifically, we generate 1k sketches per class by conducting classifier guidance on the obtained unconditional DDPM under different configurations. We have further clarified this in Appendix B in our revised version.

---

> > ### Comment · Reviewer_bpbS · 2023-11-22
> > **Q2. Also about the SGD process.**
> >
> > Q2. Also about the SGD process. In equation 4,"N is the number of sketches generated within a sampling batch". But why should the sketches in the same batch share the same guidance scale s?
> >
> > Futher question: Your answer of speeding up training seems questionable. So have you tried to use batchsize=1? And and how about the result, would it improve? Does the batchsize N affect the results?

---

> > > ### Author Response · Authors · 2023-11-23
> > > **Further clarification**
> > >
> > > Thank you for your further questions! We didn’t quite look into the choice of batchsize=1 before. But with a quick deeper investigation, we indeed found something interesting under your suggestion!
> > >
> > > When setting batchsize=1, other than the known fact that this choice can highly slow down the sampling process (this is the main reason that hinders us proceeding with it before), we further found that it makes the sampling sometimes unstable due to gradient vanishing during scale optimization, i.e., the scale gradient becomes zero (achieving local minimum). In contrast, our default setting, i.e., bs=128, offers alleviation to the issue since larger batches provide more samples for gradient computation thus slowing down the vanishing of gradients.
> > >
> > > To further inspect the robustness of our model under various choices of batchsize, and how the batchsize impact the results, we tried setting the batchsize smaller, i.e., bs=16 and bs=64. As shown in the following table, the FID score remains relatively stable while the recall increases clearly when the batchsize becomes smaller (but much more time-consuming). The trend of generation results can conform to the expectation, i.e., using smaller batchsize offers improved results.
> > >
> > > To sum up, bs=1 should be ideal for scale optimization theoretically, however it is sub-optimal due to the issue of scale gradient vanishing in practice. As a result, we altered to a batch version scale for speeding up and stabilizing the generation.
> > >
> > > We will provide a separate section in the appendix to thoroughly investigate this issue in the final version.
> > >
> > > batchsize | FID | Prec | Rec
> > > |:---:|:---:|:---:|:---:|
> > > 16 | 3.13  | 0.68 | 0.40
> > > 64 | 3.11  |0.68 | 0.38
> > > 128 | 3.08 | 0.68 | 0.35

---

### Official Review · Reviewer_st3k · 2023-11-01

**Soundness:** 2 fair
**Presentation:** 3 good
**Contribution:** 2 fair
**Rating:** 5
**Confidence:** 4

**Summary:**

The paper presents a method for class-guided sketch synthesis. The base model is an unconditional DDIM model operating in pixel space. During testing, the method uses classifier guidance to generate class-guided sketches. Naively using the same scale for all the time steps often produces low-fidelity or over-sketching samples. To address this issue, the paper proposes the scaling indicator which is computed based on stroke complexity and recognizability. At each sampling step, the classifier guidance scale is adaptively optimized to match the residual sketch with the scaling indicator. All the experiments are done on the QuickDraw dataset.

**Strengths:**

- The results are good qualitatively.
- The idea of adaptive scale optimization is interesting.
- The paper reads well and is easy to follow.

**Weaknesses:**

The validation of the idea is lacking:
- The comparison with classifier-free guidance is missing.
- In Table 1, what's the classifier guidance scale of DDIM? To validate the idea, it will be good to sweep over all possible classifier guidance scales and show that the proposed method works better than any constant guidance scale.
- Why not just replace the classifier score in Equation-1 with the scaling indicator? In this case, the residual sketch and scale optimization are not needed anymore.

**Questions:**

What's the effect of \alpha and \beta?

---

> ### Author Response · Authors · 2023-11-22
> **Response to Reviewer st3k**
>
> Thank you for your insightful comments! We respond to your specific questions below. Hope our replies and revisions have addressed your concerns.
> > The comparison with classifier-free guidance is missing.
>
> Thanks! We have now included the classifier-free diffusion guidance (CFDG) results in Table 1 in the revision. We can see that our model can also achieve better sample quality than CFDG, i.e., FID score 3.08 vs 3.75 (this is the best result with the optimal guidance strength $\omega = 2$). Note: due to limited time and compute, we can only provide results on 30 categories for now. Results of the full 345 categories will be added in the final version.
>
>
> > In Table 1, what's the classifier guidance scale of DDIM? To validate the idea, it will be good to sweep over all possible classifier guidance scales and show that the proposed method works better than any constant guidance scale.
>
> Got it. In Table 1, the scale of classifier guidance used for DDIM was set to 0.4, which is the optimal constant scale determined by greedy search. We here further show DDIM results with some alternative choices of scale. We can see that our model can consistently outperform DDIM with various constant scales. We have also clarified how the DDIM guidance scale is selected in the revised paper.
>
> ||  FID | Prec | Rec |
> |--|:--:|:--:|--|
> |DDIM (s=0.3)| 4.21 | 0.66  | 0.35 |
> |DDIM (s=0.4, *our default*)| 4.08 | 0.71 | 0.30 |
> |DDIM (s=0.5)| 4.13  |  0.68 | 0.27 |
> |DDIM (s=0.6)| 4.37 | 0.68  | 0.21 |
> |Ours| 3.08 | 0.68  | 0.35 |
>
>
> > Why not just replace the classifier score in Equation-1 with the scaling indicator? In this case, the residual sketch and scale optimization are not needed anymore.
>
> Interesting point. We tried, however this did not work. Results obtained mostly contain sparse dots or completely blank. The reason, we think, is that when using only the scaling indicator but no classifier gradient as a part of the guidance, there will be no control towards the desired category. Moreover,  the *new* guidance will corrupt the estimated noise at each step.
>
> > What's the effect of \alpha and \beta?
>
> As described in the manuscript (penultimate line page 4) , $\alpha$ and $\beta$ are used to balance the impacts between the complexity $c(x_{0|t})$ and recognizability $f(x_{0|t})$. To gain more insights into how $\alpha$ and $\beta$ impact the generation results, we provide ablative results with different settings in Figure 7(a) of Appendix B.

---

### Author Response · Authors · 2023-11-22
**General Response**

We thank all reviewers for their valuable feedback!

We have made revisions to the paper as outlined below:

- results of classifier-free guided diffusion model are added in Table 1 (reviewer **st3k**)
- parameters selection $\alpha$, $\beta$ and $\gamma$ now included in Appendix B (reviewer **st3k**, **bpbS** and **SPRZ**)
- numerical proof of convergence of $L_t(s)$ is added in Appendix C (reviewer **bpbS**)
- discussions about why we adopt a shared scale $s$ within a batch are added in implementation details (reviewer **bpbS** and **SPRZ**)
- clarifications and quantitative results about why the noisy-sketch trained classifier can be used to recognize $x_{0|t}$ are added in Appendix D (reviewer **t4As**)

---
We below made our best efforts to clarify specific questions for each reviewer.

---

### Meta-Review · Area_Chair_wkAt · 2023-12-06

**Metareview:**

The paper tackles class-guided sketch synthesis. The paper proposes a scaling indicator computed based on the complexity and recognizability of strokes. The proposed framework can achieve sketch diversity and quality, and can strike a balance between complexity and recognizability. The paper got three positive and one negative ratings. Reviewers all agree that the paper is well-written, the idea is interesting, and the visual results are good. The initial concerns included the absence of some math proof, parameters selection, and additional cfg results. These concerns were addressed during the rebuttal. I suggest authors to include the suggestions and new results into the main paper.

**Justification For Why Not Higher Score:**

The proposed idea is effective and the results are convincing. However, the scope of design choices and the mathematical soundness are not thoroughly discussed and explored, given the discussions. Furthermore, the future extension and the generalizability of this work are also unclear.

**Justification For Why Not Lower Score:**

The proposed method is effective and and claims are corroborated quantitatively and qualitatively.

---

### Decision · Program_Chairs · 2024-01-16

Accept (poster)

---

> ### Public Comment · ~Jijin_Hu1 · 2024-06-16
>
> the PyTorch implementation of the work can be found at https://github.com/HuJijin/Adaptive_Guided_Sketch